# Loss of Skeletal Muscle Inositol Polyphosphate Multikinase Disrupts Glucose Regulation and Limits Exercise Capacity

**DOI:** 10.3390/ijms26062395

**Published:** 2025-03-07

**Authors:** Ji-Hyun Lee, Ik-Rak Jung, Becky Tu-Sekine, Sunghee Jin, Frederick Anokye-Danso, Rexford S. Ahima, Sangwon F. Kim

**Affiliations:** Department of Medicine, Division of Endocrinology, Diabetes, and Metabolism, Johns Hopkins University, Baltimore, MD 21218, USA; jlee835@jh.edu (J.-H.L.); ijung4@jhmi.edu (I.-R.J.); btusekine@jhmi.edu (B.T.-S.); fadanso@jhmi.edu (F.A.-D.)

**Keywords:** inositol polyphosphate, IPMK, skeletal muscle, exercise, insulin

## Abstract

Inositol phosphates are critical signaling messengers involved in a wide range of biological pathways, and inositol polyphosphate multikinase (IPMK) functions as a rate-limiting enzyme for inositol polyphosphate metabolism. IPMK has been implicated in cellular metabolism, but its function at the systemic level is still poorly understood. Since skeletal muscle is a major contributor to energy homeostasis, we have developed a mouse model in which skeletal muscle IPMK is specifically deleted and examined how a loss of IPMK affects whole-body metabolism. Here, we report that skeletal-muscle-specific IPMK knockout mice exhibited a ~12% increase in body weight compared to WT controls (*p* < 0.05). These mice also showed a significantly impaired glucose tolerance, as indicated by their ~50% higher blood glucose levels during GTT. Additionally, exercise capacity was reduced by ~45% in IPMK-MKO mice, demonstrating a decline in endurance. Moreover, these metabolic alterations were accompanied by a 2.5-fold increase in skeletal muscle triglyceride accumulation, suggesting impaired lipid metabolism. Further analysis revealed that IPMK-deficient myocytes exhibited 30% lower β-oxidation rates. Thus, our results suggest that IPMK mediates whole-body metabolism by regulating muscle metabolism and may be potentially targeted for the treatment of metabolic syndromes.

## 1. Introduction

Inositol polyphosphate multikinase (IPMK) is the rate-limiting enzyme active in the synthesis of soluble inositol polyphosphates (IPs) and inositol pyrophosphates (PP-InsPs) and an important component in the recycling of phosphatidylinositol lipids (PIPs) [1,2,3,4,5,6]. The targeted disruption of various enzymes that participate in IP synthesis has shown that these molecules impact a variety of cellular processes ranging from gene transcription to cellular metabolism, and IPs are now fully recognized as critical second messengers [7,8,9,10,11,12]. IPK2, the yeast homolog of mammalian IPMK, was originally identified as a factor (Arg82) necessary for the assembly of the ArgR/Mcm1 complex, which reciprocally regulates the expression of the anabolic and catabolic genes required for arginine metabolism, exerting both scaffolding and activity-dependent functions [13,14]. Murine IPMK also participates in metabolic functions, including amino acid sensing and glucose homeostasis, through protein–protein interactions (PPIs) with AMPK, LKB1, and mTORC1. In addition, IPMK plays a critical function in regulating growth factor signaling, including insulin response [7,15,16,17,18].

Increased skeletal muscle mass and physical activity are beneficial for metabolic, cardiovascular, and overall health [15,16,17]. Adult skeletal muscles are composed of various fiber types, each specialized in terms of their morphology, contractile properties, metabolism, and resistance to fatigue, enabling them to undertake specialized functions [18,19]. Skeletal muscle comprises ~40% of the body mass of mammals, is the major determinant of energy expenditure at rest and during physical activity, and plays a critical role in glucose homeostasis as the predominant site of insulin-stimulated glucose disposal and glycogen storage [20,21]. Skeletal muscle makes up ~30% of whole-body metabolism at rest and up to ~90% during maximal exercise [22].

Various signaling pathways have been linked to skeletal muscle functions [23,24,25,26]. AMPK is a serine/threonine that modulates cellular metabolism acutely via the phosphorylation of metabolic enzymes and long-term via transcriptional regulation [27,28,29,30]. AMPK is activated by a cellular energy deficit, e.g., fasting and exercise, resulting in an increased AMP/ATP ratio. Acute AMPK activation suppresses glycogen and protein synthesis and promotes glucose and fatty acid oxidation, while chronic AMPK activation induces mitochondrial biogenesis through the co-activator PGC-1α [31,32] and alters metabolic gene expression via transcription factors [27,28]. It is possible to speculate that the extent to which IPMK modulates the interplay between insulin signaling and lipid metabolism in skeletal muscle is unclear based on its known interactions with key metabolic regulators such as Akt, AMPK, and mTOR [4,7,33,34]. To elucidate this role, we investigated whether skeletal muscle IPMK is a critical regulator of whole-body metabolism. Here, we report that the loss of skeletal muscle IPMK (henceforth referred to as MKO) disrupts insulin sensitivity and lipid utilization, leading to increased body weight, glucose intolerance, and reduced exercise tolerance. This study provides the first evidence that IPMK regulates skeletal muscle energy metabolism through insulin signaling and lipid utilization, highlighting its potential role in the pathogenesis of metabolic disease.

## 2. Results

A loss of IPMK in skeletal muscle disrupts energy homeostasis. To better understand the metabolic role of IPMK, we have created skeletal-muscle-specific Ipmk-null mice by crossing *Ipmk*-loxp (neomycin-deleted) mice with MLC-Cre. MKO mice were born without any obvious gross abnormalities but started to gain more weight at 13–14 weeks (Figure 1A). To assess their energy homeostasis, MKO and WT mice were housed singly in metabolic cages. The respiratory exchange ratio (RER) in MKO mice was significantly elevated, especially in the light period, compared to WT mice, indicating an impairment of fat oxidation (Figure 1B). The spontaneous locomotor activity measured using photobeam breaks was not different between MKO and WT mice (Figure 1C). The energy expenditure against lean mass was decreased in MKO mice (Figure 1D). These mice were further examined for their body composition at 22 weeks. We found that the MKO mice gained more weight and exhibited an increased fat mass compared to WT mice. The lean mass in MKO mice was slightly decreased, without statistical significance (Figure 1E,F). However, the expression of myosin heavy chain 2 (MYH2) remained unchanged in the tibialis anterior (TA) and gastrocnemius (GA) muscles between WT and MKO mice (Appendix A) Overall, these results indicated that MKO mice had a reduced oxidative capacity with lower energy availability, ultimately leading to weight gain.

Glucose homeostasis is impaired in MKO mice. Glucose is one of the main energy sources for the skeletal muscle through glycolysis and oxidative phosphorylation, and impaired glucose metabolism in muscles leads to reduced energy usage and increased fat storage, contributing to weight gain and insulin resistance [16,34]. To study the effect of an IPMK deletion in muscle on glucose homeostasis, we performed an intraperitoneal glucose tolerance test at 22 weeks of age. After 6 h of fasting, the blood glucose levels in MKO mice were significantly higher than those in WT mice (Figure 2A). Moreover, we observed that MKO mice developed a markedly impaired glucose tolerance compared to WT mice, suggesting that a loss of IPMK in skeletal muscle disrupts whole-body glucose metabolism (Figure 2B). To further investigate the metabolic consequences of this loss, we examined the phosphorylation status of AMP-activated protein kinase (p-AMPK), a master regulator of cellular metabolism and lipid homeostasis, and ribosomal protein S6 phosphorylation (p-S6), a downstream target of mTOR signaling. We found that the expression of p-AMPK and p-S6 was slightly decreased in MKO skeletal muscle (Figure 2C).

The deletion of skeletal muscle IPMK disrupts the gene expression associated with lipid metabolism. Skeletal muscle also utilizes lipids as a critical energy source via beta-oxidation. Disrupted lipid metabolism in skeletal muscle leads to fat accumulation, insulin resistance, impaired energy production, and inflammation, contributing to metabolic disorders like obesity and type 2 diabetes [16,35]. Hence, we examined the expression of genes related to lipid metabolism in gastrocnemius muscle. We noted that the expression of genes related to lipid uptake (CD36) and synthesis (SCD1) was significantly increased, while the expression of the fat oxidation gene (CPT1b) was decreased in MKO mice (Figure 3A). These changes were further associated with an increase in triglyceride (TG) in the muscle tissue of MKO mice (Figure 3B). These data suggest that a loss of skeletal muscle IPMK disrupts how the muscle utilizes lipids as an energy source.

The deletion of IPMK in skeletal muscle reduces exercise tolerance. Our data showed that the deletion of IPMK interferes with glucose and lipid homeostasis and, in particular, lipid utilization in the skeletal muscle. Thus, we proceeded to determine how the disrupted glucose and lipid metabolism in the skeletal muscle in MKO mice affects their muscle function by conducting a forced treadmill exercise with 14–15-week-old WT and MKO mice, which was before their body weights significantly diverged. The treadmill test results showed that MKO mice had a significantly shorter running time until exhaustion than WT mice, suggesting that the IPMK deficiency in their muscle impaired their aerobic exercise capacity (Figure 4).

A loss of IPMK impairs myocellular glucose utilization and insulin signaling. Previously, we have shown that IPMK is necessary for the glucose-mediated modulation of AMPK in various cell lines [7,33]. The ubiquitously expressed AMPK senses the energy status of cells, regulates fuel availability, and plays a major role in regulating glucose and lipids [27,28,29,30]. To further investigate the role of IPMK in glucose homeostasis in muscle via its interaction with AMPK, we isolated primary myoblasts from WT and MKO mice and differentiated them into myotubes. We found that in response to high glucose, the loss of IPMK attenuates glucose-mediated pAMPK inactivation in myocytes (Figure 5A). Moreover, we observed that insulin-mediated signaling events generated by the phosphorylation of Akt were inhibited in *Ipmk*^−/−^ primary myocytes (Figure 5B).

A loss of IPMK disrupts lipid homeostasis in primary myocytes. To further explore the metabolic role of IPMK, we examined the glucose uptake, triglyceride (TG) levels, fatty acid uptake, and β-oxidation in primary myocytes with and without IPMK. Both basal and insulin-stimulated glucose uptake were significantly reduced in *Ipmk*^−/−^ myocytes compared to WT ones, indicating a defect in their glucose utilization (Figure 6A). Simultaneously, TG levels were significantly increased in *Ipmk*^−/−^ myocytes compared to WT cells (Figure 6B). Moreover, we observed an increase in ^14^C-oleic acid (OA) uptake (Figure 6C) and a significant reduction in β-oxidation, which was measured by the release of ^14^CO_2_ into the media (Figure 6D). These findings suggest that the deletion of IPMK disrupts glucose utilization and lipid homeostasis in myocytes.

## 3. Discussion

High levels of intramuscular triglyceride (IMTG) are common in obese or T2D individuals and are associated with insulin resistance [36,37]. Additionally, it has been proposed that decreased fatty acid oxidation in obese individuals contributes to the accumulation of IMTG [38,39,40]. Consistent with these studies, we discovered that the depletion of IPMK led to compromised insulin signaling, elevated fat accumulation, and reduced fat oxidation in myocytes. More importantly, skeletal-muscle-specific IPMK-deficient mice exhibit a reduced exercise capacity and increased body weight. The whole-body energy homeostasis analysis of IPMK-deficient mice suggests that the weight gain in IPMK-deficient mice likely is linked to their decreased EE. Abnormal fat accumulation and the reduction in the function of endurance muscle in IPMK-deficient muscle resemble insulin resistance-induced muscle weakness. Therefore, our study highlights the role of IPMK as a key regulator of the oxidative metabolism and exercise capacity of skeletal muscle.

Skeletal muscle is the primary tissue responsible for insulin-responsive glucose tolerance and lipid oxidation, making the maintenance of its functioning mass a crucial and dynamic process [31,41]. Patients with T2DM and insulin resistance often exhibit reduced responsiveness to the insulin stimulation of glucose uptake in skeletal muscle, as well as a poor exercise performance and increased fat accumulation [42,43,44]. Additionally, many studies have demonstrated that diet-induced or genetic insulin resistance, specifically in the muscle of mice, results in the development of glucose intolerance associated with elevated circulating triglycerides [45,46]. The overexpression of kinase-dead insulin receptor (IR) or IGF-1 receptor (IGF1R) in muscle leads to glucose intolerance, high levels of circulating triglycerides, and insulin resistance in mice [47,48]. Our previous studies have shown that the deletion of IPMK reduced the activity of insulin signaling in hepatocytes and exacerbated high fat-induced insulin resistance in mice [33,49]. Consistent with these findings, the depletion of IPMK in primary myocytes led to reduced AKT activation in response to insulin and IPMK-MKO mice developing glucose intolerance, which are similar to the findings in mice with an overexpression of kinase-dead insulin receptor (IR) or IGF-1 receptor (IGF1R). The whole-body metabolism analysis of IPMK-MKO mice suggests that the skeletal muscle depletion of IPMK results in a tendency toward a decrease in total energy expenditure (Figure 1) consistent with increased fat mass and unchanged locomotion (Figure 1). Additionally, the respiratory exchange ratio in MKO mice demonstrates their significantly altered fatty acid utilization during the light phase compared to WT mice. Taken together, these data indicate that an IPMK deficiency in muscle alters insulin signaling, whole-body energy expenditure, and fuel utilization.

Previously, we showed that IPMK interacts with AMPK [7,50]. AMPK is a key sensor of cellular energy status and is activated in response to low energy levels, specifically when the AMP/ATP ratio increases [23,27]. AMPK activation enhances lipid oxidation and reduces the incorporation of fatty acids into TGs by phosphorylating the ACC in skeletal muscle [23,51,52]. Conversely, the ablation of AMPK decreases fatty acid oxidation, increases lipid accumulation in skeletal muscle, and leads to elevated TG contents [53]. Similarly, the reduction in AMPK activation in *Ipmk*^−/−^ myocytes results in decreased glucose uptake (Figure 6) and increased triglyceride accumulation. (Figure 6), mirroring the phenotypes observed in AMPK-deficient muscles. Furthermore, IPMK-deficient myocytes exhibit an increased expression of lipogenic genes, including CD36 and SCD1, while their expression of oxidative genes, such as CPT1b, is decreased. These results recapitulate the phenotypes of the AMPK-deficient mouse model, suggesting that the IPMK-AMPK axis may play a major role in the metabolic processes and functions of muscle. IPMK likely functions as an upstream regulator of the AMPK pathway, influencing both lipid and glucose metabolism. Reduced AMPK activation in IPMK-deficient myocytes likely shifts the metabolic balance toward triglyceride accumulation and diminished fatty acid oxidation, contributing to impairments in energy efficiency and exercise capacity. Furthermore, given AMPK’s well-established role in promoting mitochondrial function, an IPMK deficiency may indirectly compromise the mitochondrial oxidative capacity, exacerbating the metabolic inefficiencies observed in MKO mice. IPMK may also regulate systemic metabolism by influencing other key pathways, particularly insulin signaling. The observed reduction in AKT activation in IPMK-deficient myocytes suggests potential crosstalk between IPMK, AMPK, and insulin signaling, which together orchestrate the utilization and storage of energy in skeletal muscle. Future studies exploring these molecular interactions, including the potential effects on transcriptional regulators such as PGC-1α, will be crucial for elucidating the precise role of IPMK in skeletal muscle metabolism. These insights could provide a deeper understanding of potential therapeutic targets for metabolic syndromes.

Beyond its catalytic functions, IPMK also acts as a transcriptional coactivator, regulating immediate early gene expression and cellular responses. Xu et al. demonstrated that IPMK directly interacts with p53, enhancing its transcriptional activity and promoting apoptosis, independent of its kinase function [54]. Additionally, IPMK binds to the histone acetyltransferase CBP/p300, facilitating the transcriptional activation of CREB-regulated genes critical for memory formation [55]. These findings suggest that IPMK’s non-catalytic interactions with transcription factors and coactivators may extend beyond neuronal and apoptotic pathways, potentially influencing the gene expression networks in skeletal muscle. Moreover, IPMK has been identified as a transcriptional coactivator of SRF, a major regulator of cytoskeletal organization and muscle adaptation [56]. Because SRF interacts with Elk-1, IPMK deletion may disrupt the formation of SRF-Elk-1 complexes, potentially altering the transcription of immediate early genes that respond to growth and stress stimuli [2]. This raises the possibility that the metabolic dysfunction observed in IPMK-MKO mice may, in part, be influenced by transcriptional changes mediated by Elk-1 dysregulation. Future studies should investigate whether IPMK deletion directly impacts Elk-1 activity in skeletal muscle and how this contributes to the metabolic phenotypes observed, including insulin resistance and altered lipid metabolism. Additionally, IPMK deletion has been reported to elevate glutathione (GSH) and cysteine levels, leading to increased resistance to oxidative stress in murine embryonic fibroblast cells and liver tissue [57]. Glutathione plays a critical role in neutralizing reactive oxygen species (ROS) and maintaining the cellular redox balance. While skeletal muscle function relies on a delicate balance between ROS production and antioxidant defense, excessive antioxidant activity may impair redox-sensitive metabolic signaling pathways. With metabolic dysfunction evident in IPMK-MKO mice, including their reduced energy expenditure and altered lipid utilization, it is possible that elevated glutathione levels contribute to these phenotypes by disrupting normal redox regulation. Nevertheless, future studies should explore the role of IPMK in the skeletal muscle redox balance and its potential implications for metabolic adaptation under physiological and pathological conditions. IPMK deficiency leads to reduced oxidation and triglyceride accumulation in myocytes. Fat accumulation, along with insulin resistance, encourages the development of defects in fatty acid metabolism. These defects may include alterations in fatty acid oxidation and uptake, TG synthesis, or any combination of these processes in skeletal muscles. A reduction in oxidative enzyme activity and an increase in glycolytic activity in muscle cells indicate a shift from oxidative (aerobic) metabolism to glycolytic (anaerobic) metabolism, resulting in a decreased capacity for aerobic energy production. This shift can lead to reduced endurance [58,59,60]. The ability of IPMK to regulate the metabolic capacity of skeletal muscle may be enabled, at least in part, by its effects on genes involved in oxidative metabolism, such as SCD1, CPT1, and CD36,in response to various environmental conditions like age or exercise. Given that a decreased EE and exercise capacity were already observed in IPMK KO mice (Figure 5 and Figure 6), it is likely that altered muscle function could be attributed to the disrupted energy metabolism in IPMK-deficient mice. Moreover, it is prudent to point out that the observed reduction in endurance does not translate into heart dysfunction, as MLC-Cre specifically influenced skeletal muscle, not cardiac muscle [61].

In summary, our study demonstrates for the first time that the depletion of IPMK in skeletal muscle disrupts glucose homeostasis and increases lipid accumulation through impaired insulin signaling. Additionally, our findings reveal that IPMK plays a significant role in regulating the oxidative metabolism and exercise capacity of muscle. These results suggest a plausible mechanism for how muscle’s metabolic function may affect metabolic diseases and/or exercise capacity.

## 4. Materials and Methods

Mice. All animal experimental procedures were approved by the Johns Hopkins University Animal Care and Use Committee. To generate skeletal-muscle-specific IPMK knockout mice, *Ipmk*-floxed mice (henceforth referred to as WT) were mated with MLC-Cre transgenic mice (a generous gift from Dr. Se-Jin Lee, University of Connecticut) (MKO). *Ipmk*-floxed mice were developed, as described previously [33] and backcrossed with a C57BL/6J wild type for at least five generations. All mice were maintained in a 12:12 h light–dark cycle with free access to regular chow (4.5% fat, 49.9% carbohydrate, 23.4% *protein*; 4 kcal/g, catalog no. 5001, LabDiet, Richmond, IN, USA) and water in a specifically pathogen-free facility in the Johns Hopkins University.

Primary myocytes. Skeletal muscle tissues were isolated from 3 to 5 neonate (1–3 days old) pups and minced with a blade. The tissues were mixed with 2 mL of Collagenase/Dipase/CaCl_2_ solution added per gram and incubated at 37 °C for 20 min. Myocytes were collected by centrifugation at 350× *g* for 5 min and resuspended in F-10-based primary myoblast growth medium (Ham’s F-10 nutrient mixture: 20% fetal calf serum, 0.5% bovine serum, and 1% penicillin/streptomycin (100 U/mL and 100 ug/mL each) in a tissue culture dish coated with collagen. They were incubated at 37 °C in a 5% CO_2_ incubator and the medium was changed every 2 days. Once fibroblasts were no longer visible, the medium was switched to F-10/DMEM-based primary myoblast growth medium (F-10:DMEM = 1:1) For differentiation, the medium was replaced with Fusion Medium (DMEM with 5% horse serum and penicillin/streptomycin).

Measurement of body composition, energy balance, and indirect colorimetry. The body composition of the mice was measured by an EcoMRI (dual-energy X-ray absorptiometry) scan. Their energy expenditure was measured by open-circuit calorimetry (Oxymax system, Columbus Instruments, Columbus, OH, USA) and their locomotor activity was simultaneously measured by infrared beam interruption (Opto-Varimex System, Columbus Instruments, OH). Mice were housed individually in calorimetry cages at 23 °C. Room air was pumped in at a rate of 0.52 L/min and exhaust air was sampled at 15 min intervals for 5 h. Their oxygen consumption (VO_2_) and carbon dioxide production (VCO_2_) were measured using electrochemical and spectrophotometric sensors, respectively. The respiratory quotient (RQ), a measure of fuel use, was calculated as the ratio of oxygen consumption to carbon dioxide production. A decrease in RQ indicates fatty acid oxidation. Total energy expenditure (heat) = Calorific value (CV) × VO_2_, where CV = 3.815 + 1.232 × RQ.

Treadmill exercise. For the assessment of their muscle endurance capacity, WT and IPMK MKO male mice (n = 8 per group) were tested using treadmill running. For the treadmill running, an Open Rodent Treadmill Exer-3/6 was used. Prior to exercise, mice became accustomed to the treadmill with a 2–3 min exploration once per day for 3 days. The exercise test was performed on a 10% incline at 10 m/min for 5 mins, followed by an increase toward 30 m/min until exhaustion.

Glucose tolerance test. Mice were fasted for 6 h and glucose (2 g/kg body weight; i.p.) was administered intraperitoneally. After injection, blood was taken by puncturing the tail vein, and their glucose levels were measured using a glucometer (Ascensia Diabetes Care, Parsippany, NJ, USA) at 0, 15, 30, 60 and 120 min. Blood glucose was also measured before the injection (time point 0).

Quantitative real-time PCR. Total RNAs were isolated using TRIzol Reagents. RNAs were reverse-transcribed into cDNA using Protoscript II (New England Biolabs, Ipswich, MA, USA). In total, 1–2 μg of cDNA was diluted to 2 ng/mL and was amplified by specific primers in a 10 µL reaction using SYBR Green Master mix (Applied Biosystems, Carlsbad, CA, USA). The analysis of gene expressions was carried out in QuantStudio 5 (Applied Biosystems). The mouse TATA-binding protein gene (*Tbp*) was used as the reference gene, and data were normalized and relative expressions determined from the threshold cycle (Ct) following the 2^−ΔΔCT^ method. The primers used were as follows: SCD1 forward 5′-GCAAGCTCTACACCTGCCTCT-3′, reverse 5′-CGTGCCTTGTAAGTTCTGTGGC-3′; DGAT1 forward 5′-TGACCTCAGCCTTCTTCCATGAGT-3′, reverse 5′-CCACACAGCTGCATTGCCATAGTT-3′; CPT1b forward 5′-GGCACCTCTTCTGCCTTTAC-3′, reverse 5′-TTTGGGTCAAACATGCAGAT-3′; CD36 forward 5′-ACTGGTGGATGGTTTCCTAGCCTT-3′, reverse 5′-TTTCTCGCCAACTCCCAGGTACAA-3′;

FATP1 forward 5′-CCGTCTGGTCAAGGTCAATG-3′, reverse 5′-CACTAACATAACCATCGAAACGC-3′.

Western blotting: Samples were lysed in ice-cold RIPA buffer containing protease and phosphatase inhibitors and heated at 95 °C for 5 min prior to electrophoresis. Proteins were transferred to a 0.2 µm nitrocellulose membrane, which had been blocked with 5% nonfat milk in Tris-buffered saline for 30 min. The blots were then incubated with the primary antibodies at 4 °C. Immunoblotting was conducted with the following antibodies: IPMK from Novus Biologicals; pAMPK, AMPK, pAkt-S473, Akt, pS6P, S6P, and HSP90 from Cell Signaling Technology; and MYH2 from Developmental Studies Hybridoma Bank. Blots were imaged and quantitated using an Odyssey Near-Infrared Scanner (Li-Cor Biosciences, Lincoln, NE, USA).

Glucose uptake Assay. Cells were washed with PBS and incubated in serum-free medium for 2 h. Then, the cells were incubated in freshly prepared Krebs-Ringer biocarbonate-Hepes buffer (KRBH, 30 mM/pH7.4 HEPES, 10 mM NaHCO_3_, 120 mM NaCl, 4 mM KH_2_PO_4_, 1 mM MgSO_2_, 1 mM CaCl_2_) with 1 mM 2-deoxyglucse plus 1μCi [14C]-2-deoxyglucose for 10 min. Cells were washed with cold PBS (×3) and their radioactivity was measured using a scintillation counter. Their glucose uptake was normalized by the protein concentration in each well.

Fatty acid uptake and oxidation. The measurement of fatty acid oxidation was performed as previously described. Briefly, primary myocytes were starved for 2 h and then incubated for 4 h in 5 mM of glucose and 0.1 mM of oleic acid pre-complexed to 0.13% BSA [1–14 C]-Oleic acid (0.1 μCi/mL). The trapped ^14^CO_2_ and ^14^C acid-soluble products were counted to estimate total oleic acid oxidation using the liquid scintillation analyzer Tri-Carb 4810 TR. The results were normalized using the cell’s protein content.

Statistical analysis. The results were analyzed using a statistics software package (GraphPad Prism version 9.0.0 for Windows, GraphPad Software, San Diego, CA, USA). We assessed statistical significance by performing unpaired two-tailed *t*-tests, a one-way ANOVA (Dunnet’s Test), or a two-way ANOVA (Tukey’s test) for multiple comparisons. The level of significance was set to *p* < 0.05.

## Figures and Tables

**Figure 1 ijms-26-02395-f001:**
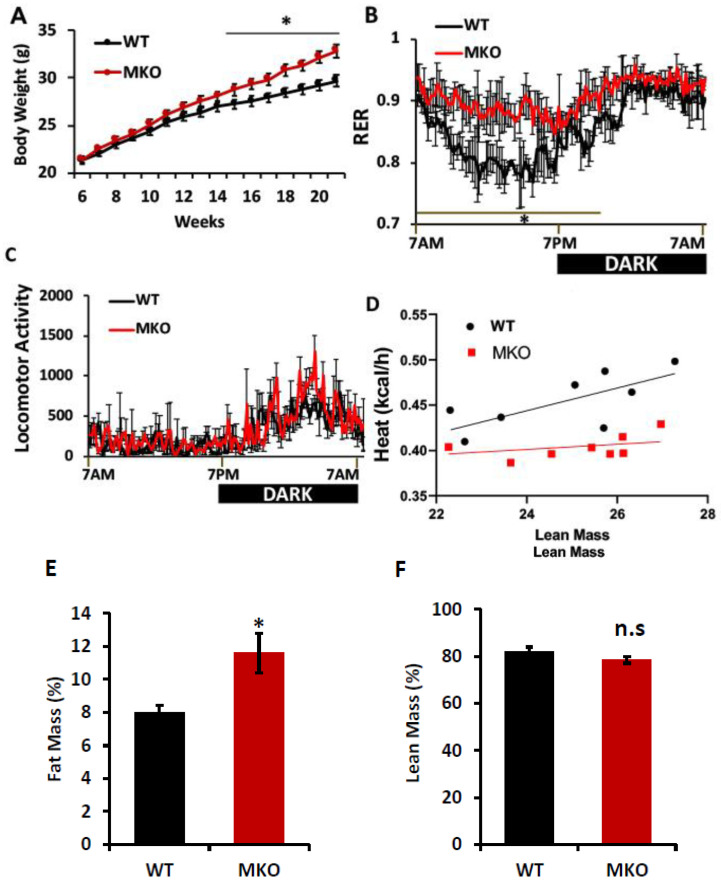
Effects of skeletal muscle IPMK deficiency on whole-body energy metabolism. Male (22–24-week-old) WT and MKO mice were examined. (**A**) Body weight, (**B**) Indirect calorimetry was performed to assess the respiratory exchange ratio (RER; VCO_2_/VO_2_), (**C**) locomotor activity measured with photobeam breaks, (**D**) regression analysis of energy expenditure, (**E**,**F**) lean and fat mass were determined by ^1^H-MRS analysis. Comparisons between the two groups were made using Student’s *t*-test. The differences between genotypes over 24 h were analyzed by a two-way ANOVA and Bonferroni post hoc test. Data are mean ± SEM, (WT, n = 8; MKO, n = 8). * *p* < 0.05; n.s., not significant.

**Figure 2 ijms-26-02395-f002:**
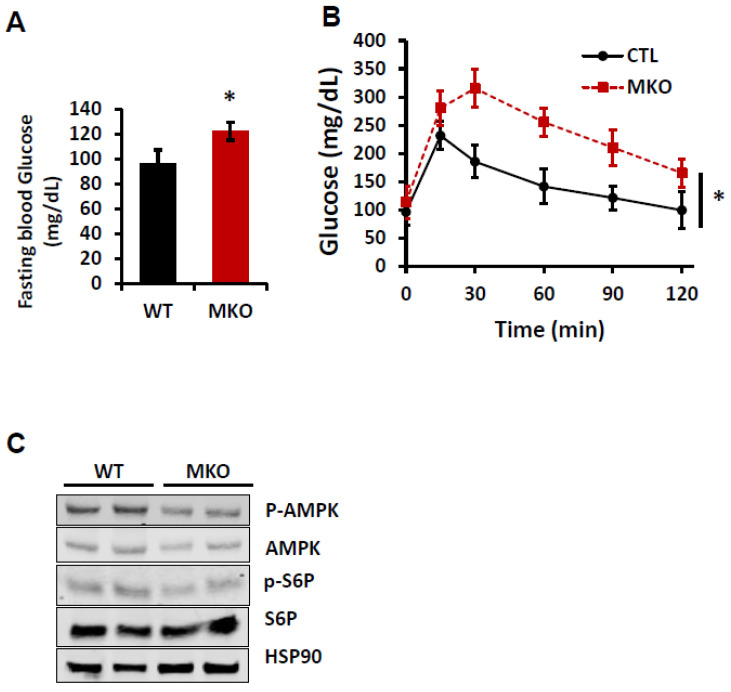
Effects of skeletal muscle IPMK deficiency on glucose homeostasis. Male (22–24-week-old) WT and MKO mice were examined. (**A**) Overnight fasting glucose and (**B**) glucose tolerance test (GTT) (n = 8 mice each group). (**C**) Expression of p-AMKP, AMPK, p-S6P, and S6P proteins in GA of WT and MKO mice (n = 2 mice each group). HSP90 protein is shown as loading control. Data are mean ± SEM, * *p* < 0.05.

**Figure 3 ijms-26-02395-f003:**
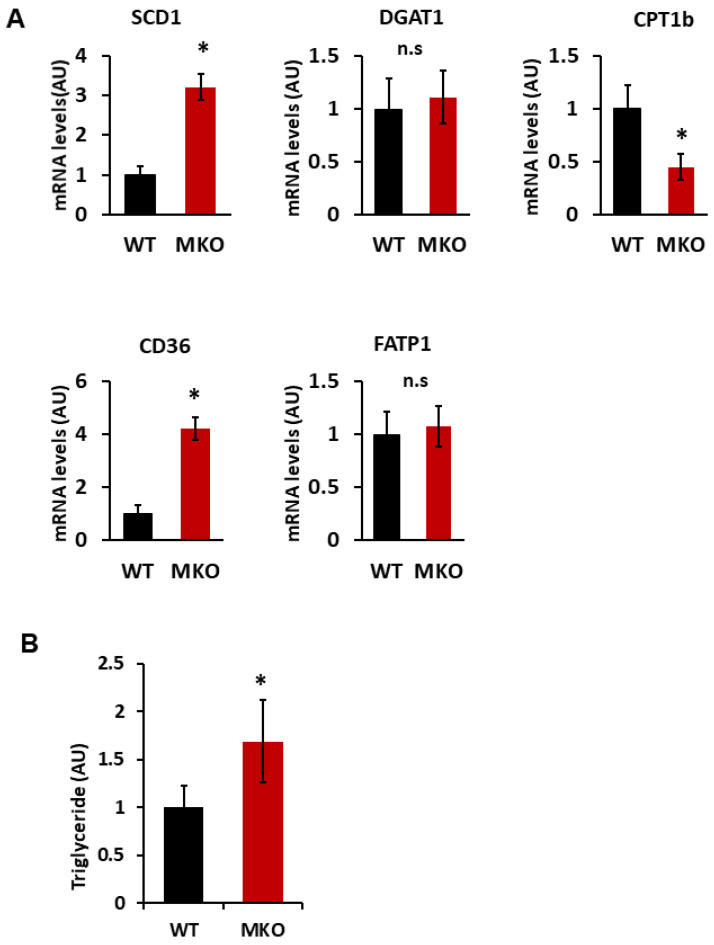
A loss of IPMK increases lipid accumulation. (**A**) Gene expression in gastrocnemius, identified by RT-qPCR (n = 6). (**B**) TG contents in the gastrocnemius muscle (n = 8). Measurement was normalized by protein concentration. Data are mean ± SEM, * *p* < 0.05; n.s., not significant. Stearoyl-CoA desaturase-1; SCD1 (a rate-limiting enzyme in the formation of monounsaturated FA). Diacylglycerol O-acyltransferase 1; DGAT1 (catalyzes the conversion of diacylglycerol and fatty acyl CoA to triacylglycerol). Carnitine palmitoyl transferase 1B; CPT1b (a rate-limiting enzyme for β-oxidation). Cluster of differentiation 36; CD36 (used in the uptake of long-chain fatty acids). Fatty acid transport protein 1; FATP1 (used in the uptake of long-chain fatty acids).

**Figure 4 ijms-26-02395-f004:**
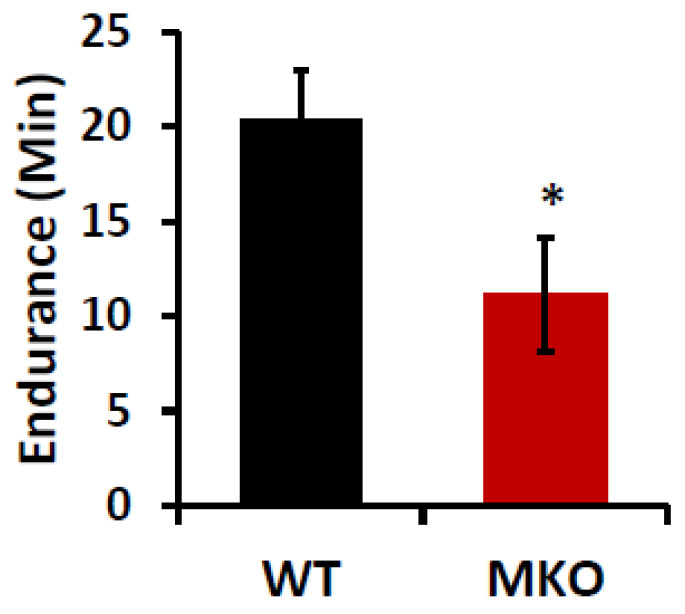
IPMK deletion in skeletal muscle reduces exercise capacity. WT and IPMK MKO male mice (14–15 weeks old) were subjected to exercise tolerance tests. Data are mean ± SEM, n = 8. * *p* < 0.05.

**Figure 5 ijms-26-02395-f005:**
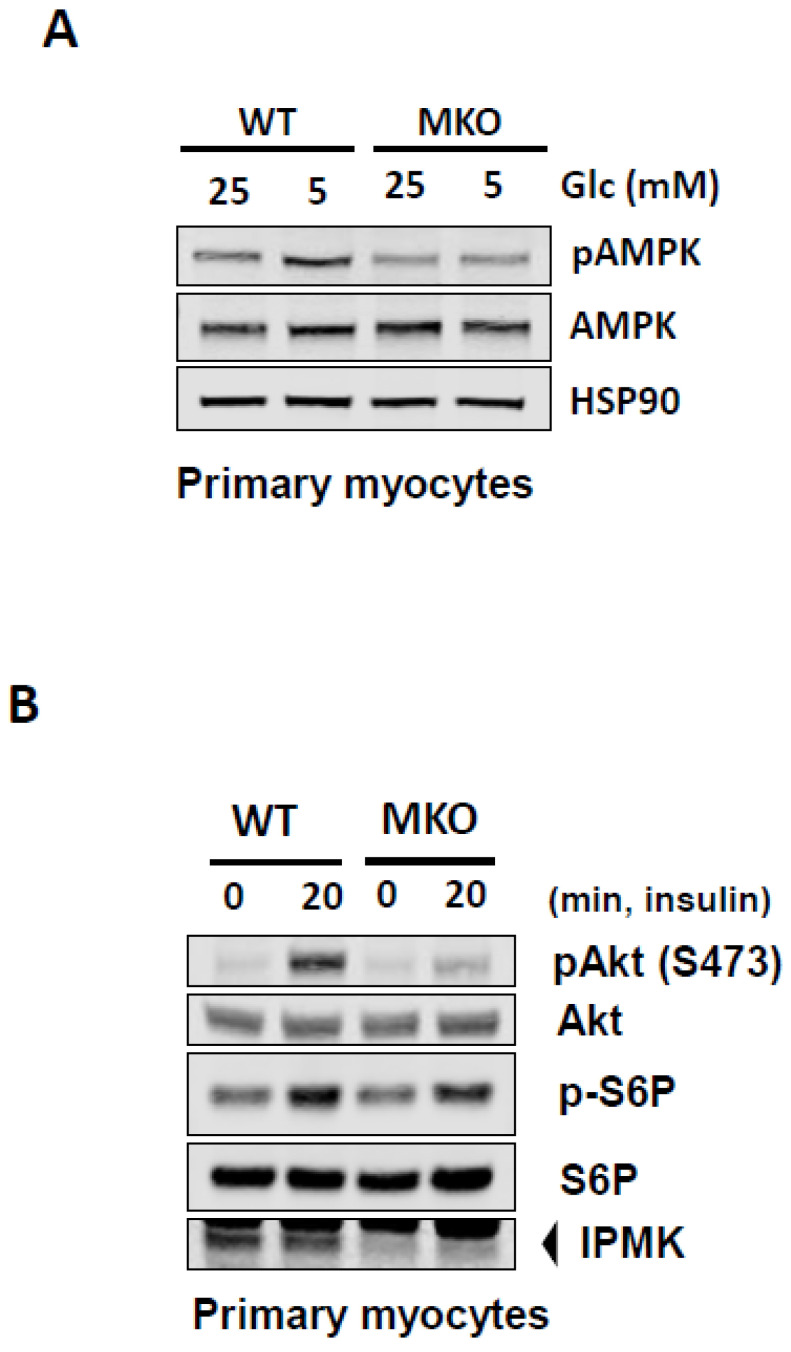
A loss of IPMK impairs myocellular glucose metabolism and insulin signaling. Primary myoblasts isolated from WT and MKO mice were differentiated into myotubes. (**A**) The cells were incubated with either 25 mM or 5 mM of glucose for 18 h. (**B**) The cells were serum-starved for 18 h and then treated with 100 nM of insulin at the indicated times. These are representative images from three independent trials of immunoblotting.

**Figure 6 ijms-26-02395-f006:**
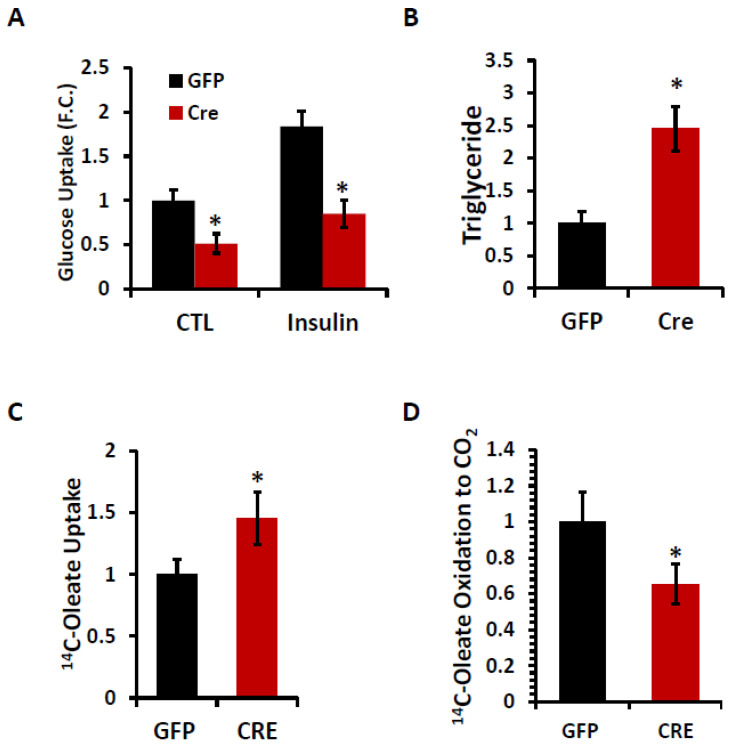
A loss of IPMK disrupts lipid homeostasis. Primary myoblasts were isolated from *Ipmk*-loxp mice and treated with either Ad-GFP as a control or Ad-Cre. Then, the cells were differentiated into myotubes. (**A**) The cells were serum-starved for 18 h and their glucose uptake was measured with or without 100 nM of insulin. (**B**) TG levels were measured. (**C**) [1-^14^C]-oleic acid uptake (normalized by the protein concentration). (**D**) The β-oxidation of [1-^14^C]-Oleic acid was measured in terms of the CO_2_ released into the media and data were normalized by the protein concentration. Data are mean ± SEM, * *p* < 0.05.

## Data Availability

The data presented in this study are openly available on bioRxiv at https://www.biorxiv.org/content/10.1101/2024.07.28.605526 (accessed on 5 August 2024).

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
