# Peer review of "Loss of Skeletal Muscle Inositol Polyphosphate Multikinase Disrupts Glucose Regulation and Limits Exercise Capacity"

_ijms, 2025, doi:10.3390/ijms26062395_

Round 1

Reviewer 1 Report

Comments and Suggestions for Authors

1. The authors provide a methods used for generating the skeletal muscle-specific IPMK knockout mice. However, the manuscript would benefit from a more detailed explanation of the controls used throughout the experiments, particularly in the context of comparing the MKO mice to wild-type. 

2. The link between IPMK deficiency, changed glucose metabolism, and exercise intolerance is well established through the presented data. However, the discussion could be expanded to speculate on the potential molecular mechanisms underlying these observations. This might include how IPMK interactions with AMPK and other metabolic regulators might influence muscle function and systemic metabolism, providing a deeper insight into the potential therapeutic targets for metabolic syndromes.

3. The figures are generally clear and support the manuscript's claims effectively. Nonetheless, I recommend enhancing Figure 3 by adding subpanels that clearly delineate the control versus experimental groups for each gene expression analysis. This would aid in quicker visual comprehension of the comparative results. 

4. I recommend also the investigation of mTOR pathways and protein synthesis as well as the expression of myosin heavy chain in muscles. 

Reviewer 2 Report

Comments and Suggestions for Authors

The authors aimed to elucidate the role of skeletal muscle IPMK in muscle energy metabolism through the regulation of insulin response and lipid utilization.

The abstract section does not have any numerical results. It may be rectified and results in the form of numerical should be added.

The last paragraph of the introduction section should have the gap in existing literature clearly described and the need to bridge such a gap with the present study. How is the present study different from past studies? The novelty of the study should be reflected in this manner.

IPMK acts non-catalytically as a transcriptional coactivator for p53, mediating its proapoptotic influences and binds CBP/p300, enhancing its transcriptional coactivation of CREB regulated genes involved in memory. These facts should be discussed.

IPMK is a transcriptional coactivator for SRF. This fact should be discussed.

Does IPMK deletion influence Elk-1 activity?

The depletion of IPMK leads to elevated glutathione and cysteine levels, resulting in increased resistance to oxidants. Could this fact be discussed in detail?

The authors should include the limitations of the study.
